# The Application of Graph in BIM/GIS Integration

Junxiang Zhu [1,*], Heap-Yih Chong [1], Hongwei Zhao [2], Jeremy Wu [1], Yi Tan [3] and Honglei Xu [4]

1. School of Design and the Built Environment, Curtin University, Bentley, WA 6102, Australia
2. Institute of Agricultural Resources and Regional Planning, Chinese Academy of Agricultural Sciences, Beijing 100081, China
3. College of Civil Engineering, Shenzhen University, Shenzhen 518060, China
4. School of Electrical Engineering, Computing and Mathematical Sciences, Curtin University, Bentley, WA 6102, Australia
* Correspondence: junxiang.zhu@curtin.edu.au

**Abstract:** Information exchange between building information modelling (BIM) and geographic information system (GIS) is problematic, especially in terms of semantic information. Graph-based technologies, such as the resource description framework (RDF) and the labelled property graph (LPG), are promising in solving this problem. These two technologies are different but have not been systematically investigated in the context of BIM/GIS integration. This paper presents our systematic investigation into these two technologies, trying to propose the proper one for BIM/GIS data integration. The main findings are as follows. (1) Both LPG-based databases and RDF-based databases can be generally considered graph databases, but an LPG-based database is considered a native graph database, while an RDF-based database is not. (2) RDF suits applications focusing more on linking data and sharing data, and (3) LPG-based graph database suits applications focusing more on data query and analysis. An LPG-based graph database is thus proposed for BIM/GIS data integration. This review can facilitate the use of graph technology in BIM/GIS integration.

**Keywords:** building information modelling (BIM); geographic information system (GIS); industry foundation classes (IFC); labelled property graph; resource description framework; interoperability; graph





## 1. Introduction

### 1.1. Overview of BIM and GIS Integration

The integration of building information modelling (BIM) and geographic information systems (GISs) enables the creation of large-scale virtual city models, which can contribute to the development of smart cities and digital twins [1], which has been investigated by researchers from the Architecture, Engineering, and Construction (AEC) domain and the geospatial industry over the past ten years, at both the application level and the fundamental data level. While application-level studies primarily investigate the joint use of BIM and GIS technologies in solving practical problems, such as flood damage assessment [2,3], traffic noise assessment [4], supply chain management [5,6], offshore platform disassembly [7], and construction risk management [8], data-level studies investigate how to effectively integrate data from these two diverse sources to better support the upper application-level studies [9]. Data-level studies are required because BIM data cannot be directly and fully read by GIS applications, which is an across-domain interoperability issue.

### 1.2. The Interoperability Issue between BIM and GIS

Interoperability issues are common between information systems, such as GISs that integrate heterogeneous data from various sources, such as satellite images from remote sensing [10], topographic data from surveying and photogrammetry [11], location data from positioning systems [12], and other textual or tabular data. An early study on interoperability is from Bishr [13], which investigated the interoperability issue among

GISs and proposed six levels of interoperability in relation to network protocols, hardware and OS (Operating System), spatial data files, DBMS (database management system), data model, and application semantics, respectively.

The interoperability can be improved by advances in the underlying technologies [13]. By far, many of the interoperability issues have been properly addressed for GIS, such as those related to network protocols, hardware and OS, spatial data files, and DBMS. For example, common database APIs (Application Programming Interface), such as ODBC (Open Database Connectivity) and JDBC (Java Database Connectivity) [14], can be used to improve interoperability at the DBMS level.

This interoperability theory can also be applied to BIM/GIS integration. Between BIM and GIS, the interoperability issue is at both the syntactic and semantic levels [15,16]. Syntactic interoperability is mainly about accessing information, i.e., how to read building information, while semantic interoperability is mainly about understanding information, i.e., how to interpret building information. Bishr [13] proposed two approaches for improving data interoperability, including (1) providing the data receiver with information about the data and (2) providing a mechanism that enables automatic data conversion. In the context of BIM/GIS integration, most studies adopted the second approach. They converted BIM data in the open IFC (Industry Foundation Classes) format into common GIS formats, such as CityGML (City Geography Markup Language) [17], GML (Geography Markup Language) [2], shapefile [18,19], 3D tiles [20], and 3D scene layer (I3S) [21]. To facilitate BIM-to-GIS data conversion, there are also efforts in developing international standards, such as the ISO/TR 23262:2021 [22] for GIS(geospatial)/BIM interoperability and the ISO/TS 19166:2021 [23] for BIM to GIS conceptual mapping (B2GM).

### 1.3. The Basic Pattern for BIM-to-GIS Data Conversion

The mainstream of BIM/GIS integration is to use building information in GIS [24,25], due to the different strengths of BIM and GIS. In the workflow of BIM/GIS integration, BIM is usually used as an information source and GIS is used for information processing and analysing, and BIM data need to be converted.

During the data conversion, two aspects of BIM data need to be considered, i.e., geometry and semantics. Geometric information is about the size, shape, location, and orientation of models, while semantic information provides additional information about these models, such as attributes, relations, and properties, which is also referred to as non-geometric information [26,27]. Due to their diverse natures, these two types of information need to be handled separately.

Accordingly, data-level integration of BIM and GIS mainly deals with two tasks, i.e., geometry conversion and semantics transfer [24,25]. Geometry conversion deals with the conversion of 'shape', so that the shapes of building components created by BIM, both explicit shapes (e.g., boundary representation, or B-rep) and implicit shapes (e.g., swept solids and constructive solid geometry), can all be converted into explicit shapes to be used in GIS [28–30]. The process of converting implicit models into explicit models is referred to as boundary evaluation [31]. Semantics transfer deals with the transfer of semantic information (or the 'information' in BIM). The separately processed geometric information and semantic information are then combined again on the GIS side and used in practice [2,24]. This is the basic pattern for BIM-to-GIS data conversion (see Figure 1). A thorough review of BIM/GIS data integration has been presented by Zhu et al. [24].

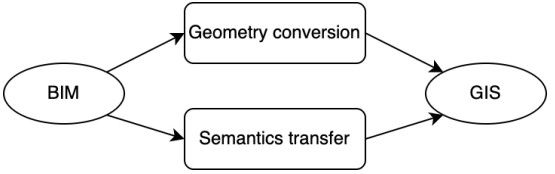

**Figure 1.** The basic pattern for BIM-to-GIS data conversion.

### 1.4. Problems in BIM-to-GIS Conversion Paths

This basic conversion pattern can be expanded into detailed paths for converting BIM data into different formats on the GIS side. Zhu et al. [24] have identified four conversion paths on top of this basic pattern, including the dataset-oriented path, the system-oriented path, the data-model-based path, and the ontology-based path. Unfortunately, these conversion paths cannot address the interoperability issue on their own.

First, the dataset-oriented path has problems in geometry conversion and semantics transfer. (a) For the IFC-to-CityGML conversion, an effective and efficient way for the solid-to-surface conversion is absent, and the class mapping problem has not been well addressed. (b) For the IFC-to-shapefile conversion, the main problem is semantics transfer. The interconnected BIM information, which can be thought of as a graph according to Zhu et al. [21], cannot be effectively represented in the traditional relational database that is used by GIS [32]. (c) A relatively new format used in BIM/GIS integration is 3D Tiles, which is a format used by Cesium [33] to stream 3D geospatial datasets for web-based applications [34]. The IFC-to-3D-tiles conversion, like the IFC-to-shapefile conversion, mainly focuses on geometry and transfers limited semantics. Second, the data-model-based path mainly creates project-specific data models by identifying information requirements for projects. For example, the BO-IDM (BIM-oriented indoor data model) developed by Isikdag et al. [35] presents information requirements for indoor navigation. Using this path, only the required information was extracted and transferred. Third, the ontology-based path only focuses on semantics only, in which the geometry is neglected.

When the use of a single conversion path cannot solve the problem, a potential way to improve data interoperability is to jointly use several paths. For example, the geometric information can be handled by using shapefile or 3D Tile (the dataset-oriented path), while the semantic information can be stored in Resource Description Framework (RDF) (the ontology-based path), or other graph-based formats that can effectively represent the relationships [26].

### 1.5. Using Graph to Improve Interoperability

Graphs can be used to represent interconnected information [36], such as BIM information, and support efficient and effective information retrieval and analysis by using graph-traversal based algorithms [32]. Due to the flexible structure and the high performance in information storage and retrieval, graph databases are being adopted by many areas, such as the AEC domain [37,38], the geospatial industry [39], and the biomedical domain [40,41].

In a general sense, a graph can model most things in the world [42], and according to [43], a graph-based model is in many cases likely to facilitate the integration of information. In this sense, in the context of BIM/GIS integration, there would be a new form of using building information on the GIS side, which is 'geometry plus graph' (see Figure 2), instead of the traditional 'geometry plus tables'.

### 1.6. Motivation, Aim, and Objectives

RDF is the commonly investigated technology for representing BIM information, and another emerging technology for that purpose is the labelled property graph (LPG) based graph database, such as neo4j [44]. A preliminary literature review shows that both RDF and LPG-based graph databases were involved in BIM/GIS data integration. However, many questions remain unsolved regarding these two technologies and their application in BIM/GIS integration.

First, (1) what is the relationship between RDF and graph? While W3C [45] and many studies [46,47] described RDF as a graph-based technology, there were also concerns over the graph nature of RDF [48], as graph traversal based information query was not directly supported by triple stores (RDF-based database). Second, (2) what is the relationship between RDF and LPG? Despite both RDF triple stores and LPG-based databases being considered graph databases and used to store connected data [32,49], they were referred to

as two different technologies created for different purposes in some studies [49,50]. Finally, (3) which technology is more suitable for BIM/GIS data integration, the RDF-based triple store, or the LPG-based graph database?

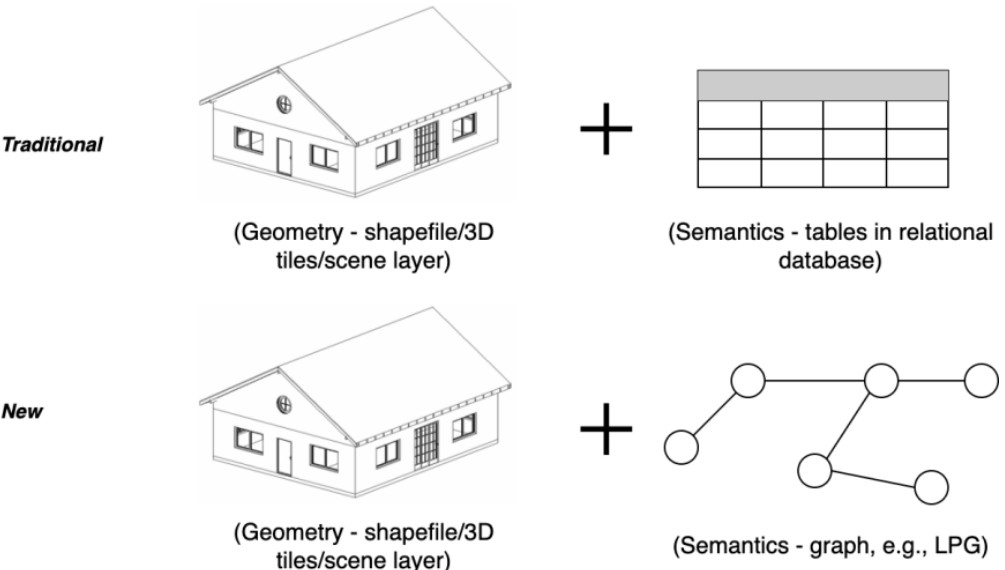

**Figure 2.** A new form of using building information in GIS by using graph.

The above questions have not been systematically investigated by previous studies. The aim of this study is thus to answer these questions by reviewing the literature. The following objectives have been identified: (1) understanding graph technology, and how graphs contribute to BIM/GIS integration; (2) understanding RDF and LPG technologies, including their initial purposes, features, as well as their functions; (3) understanding the current state of using graphs in BIM/GIS integration; (4) choosing the proper graph technology for BIM/GIS integration, and (5) identifying future work to facilitate the use of graph in BIM/GIS integration.

## 2. Methodology

To achieve the aim of this study, a literature review was conducted [51,52]. The rationales for adopting a literature review, instead of an experiment-based methodology, are as follows. (1) Currently, there is no method available for fully converting IFC into LPG. Even though there is currently a method for converting IFC into RDF, without a valid method for fully converting IFC into LPG, a comprehensive comparison cannot be carried out between RDF and LPG. (2) There are studies comparing LPG and RDF using specific use cases, where limited data were converted, but these studies observed different results and failed to explain the cause. (3) A theoretical comparison can reveal the root differences between these two technologies, which can also help the selection between these two technologies.

### 2.1. Literature Collecting and Filtering

Papers published in journals and conferences were retrieved from online databases, such as Web of Science and Scopus, using keywords 'BIM', 'GIS', and 'Graph'. Preliminary filtering was conducted to filter out papers without detailed methods for graph generation, and eventually, 36 papers were closely examined. Books on fundamental concepts of database management systems were referenced as well.

### 2.2. Literature Analysis

The collected literature was then classified into IFC-to-LPG and IFC-to-RDF and analysed accordingly. The findings are presented in the following sections with the structure

shown in Figure 3. Section 3 gives an overview of the use of graphs in BIM/GIS data integration; Section 4 focuses on the general IFC-to-Graph conversion, while Sections 5 and 6 further focus on IFC-to-LPG and IFC-to-RDF, respectively. Section 7 selects the proper graph based on the needs of BIM/GIS data integration, and Section 8 discusses future work.

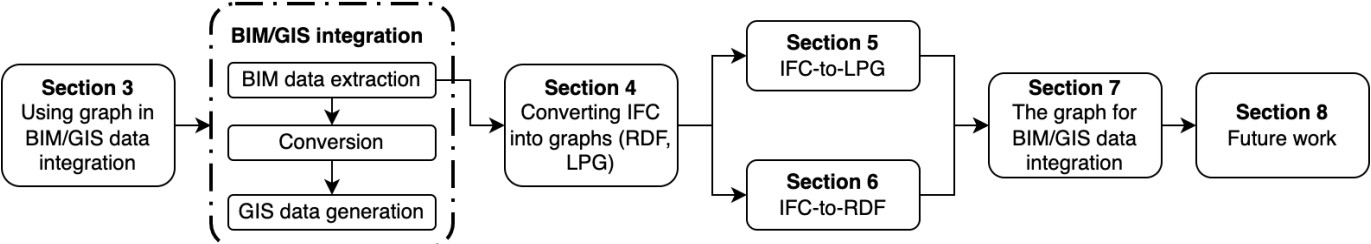

**Figure 3.** The structure of the paper.

## 3. Using Graph in BIM/GIS Data Integration

### 3.1. Graphs and Graph Theory

Graph is a term generally used to describe diagrams consisting of nodes and edges. A graph can be mathematically described by using the equation, $G = (V, E)$, where $G$ is a graph, $V$ indicates the set of vertices in the graph, and $E$ is the set of edges in the graph [53], and $E \subseteq (V \times V)$. The graph theory studies graphs, including graph type (e.g., pseudograph, directed graph, labelled graph, and multigraph), graph structure (e.g., trees, cycles, networks), features of graph structure (e.g., distance, connectivity, and planarity), and graph-based algorithms (e.g., the Dijkstra's algorithm for finding the shortest path between two nodes), as well as using these algorithms to solve practical problems. The basic concepts in graph theory include node, edge, path, connectivity, and graph traversal [54]. In general, a graph can be used to model all kinds of scenarios due to its general-purpose, and expressive structure [42].

The main applications of graphs include the following. (a) Information visualisation. Graphs are more expressive than tables and texts, and more importantly, they are more intuitive and easier to understand. For example, the IFC schema expressed in UML (Unified Modelling Language, a language for graphically representing data models/standards [55]) is easier to understand than the text-based EXPRESS schema. In this sense, a graph is also helpful in understanding the structure of datasets in many fields, such as business, government, and science. (b) Data storage. Graph-based databases, such as neo4j, can use nodes and edges to store information. Data, especially relationships, stored in a graph database can be more efficiently retrieved than from a conventional relational database [49]. (c) Data analysis. Graph theory focuses on the modelling of networks of connected elements [43], and the connections between elements can be identified through networks. Graph algorithms have been widely used in computer science, information communication, and geospatial science. For example, Dijkstra's algorithm is commonly used in pathfinding and navigation [56,57].

### 3.2. Graphs in BIM/GIS Data Integration

Graphs are being used in BIM/GIS integration, mainly at the application level, where BIM, GIS, and graphs were jointly used to solve problems such as indoor navigation and emergency response [35,58–61]. In these studies, the basic pattern is (a) extracting building information and geospatial information, (b) generating the geometric graphs (or geometric network), such as the multi-purpose geometric network model by Teo and Cho [61], and (c) applying the path finding algorithm.

At the data level, there is a smaller number of studies. These studies mainly convert BIM and GIS data into graphs, e.g., in the form of RDF triples or LPG. For example, Stouffs et al. [62] converted IFC and CityGML into graphs separately and connected them via a correlation graph, which is referred to as the Triple Graph Grammar (TGG), to convert

IFC data into CityGML. Hor et al. [32] created a semantic graph database by converting CityGML and IFC into RDF, which was later converted into a property graph in neo4j. Malinverni et al. [39] investigated the conversion of CityGML into RDF.

Even though only a limited number of studies are directly dedicated to BIM/GIS data integration using a graph, there are more studies available on converting IFC into graphs, which can contribute to the first steps of BIM/GIS data integration [24]. The whole process of BIM/GIS data integration is presented in Figure 4, as well as the potential contribution of graphs in this process. The next section is about converting IFC into graphs.

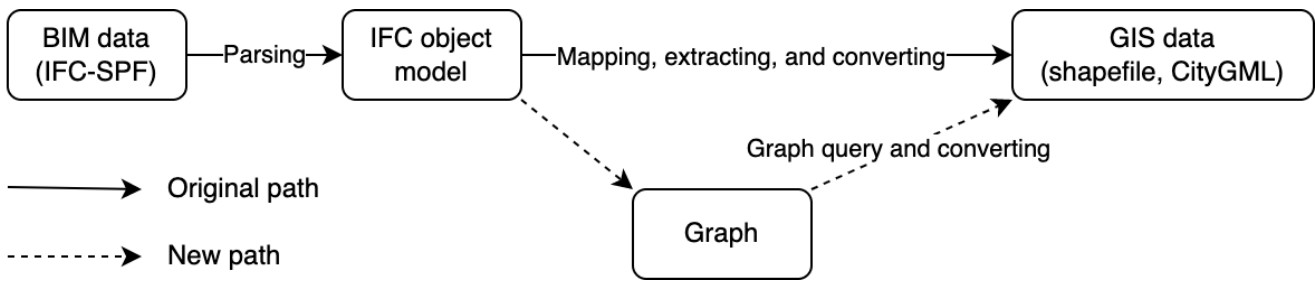

**Figure 4.** Graph contributing to BIM/GIS data integration.

## 4. Converting IFC into Graphs

### 4.1. Investigating Industry Foundation Classes (IFC)

Industry Foundation Classes (IFC) is an international standard (ISO 16739) developed by buildingSMART (formerly the International Alliance for Interoperability, or IAI) in response to the interoperability issue within the AEC domain [63]. As an open data standard, it is vendor-neutral and not specific to certain software applications. As a result, IFC is supported by most BIM authoring tools, such as Autodesk Revit, ArchiCAD, and Tekla. Due to the interoperability that IFC provides, it is dominant within the AEC domain for information exchange and BIM-related research, especially open BIM and BIM/GIS integration [24].

#### 4.1.1. IFC Formats

The IFC standard covers almost all aspects of buildings throughout the life cycle, from the early planning, design through construction, operation and maintenance, and eventually to demolition [63]. The size of IFC is thus huge but still growing [64]. The IFC schema is originally described by EXPRESS, a text-based data modelling language formalised in ISO 10303, and the classic IFC data are STEP files (IFC-SPF, or IFC STEP Physical File). To further improve the data interoperability, other formats were adopted as well, such as the XML (Extensible Markup Language) based ifcXML, the JSON (JaveScript Object Notation) based ifcJSON, and the Turtle or XML-based ifcOWL [65]. Nevertheless, the IFC-SPF is still the most used format.

#### 4.1.2. Representing IFC Data Using Graph

Figure 5a presents a portion of an IFC-SPF file. IFC-SPF uses a referencing mechanism, where instances are denoted by numbers (i.e., unique identifiers within an IFC-SPF) and can be referenced by other instances within the same file. A similar referencing mechanism is used by ifcXML and ifcJSON. So, IFC information is intrinsically interconnected and can naturally be represented by graphs. Figure 5 presents six instances (i.e., #1, #3, #7, #10, #11, and #12) and their corresponding graph representation (nodes and edges). It is obvious that the graph representation is more intuitive in revealing the relationships between instances than the text-based IFC-SPF.

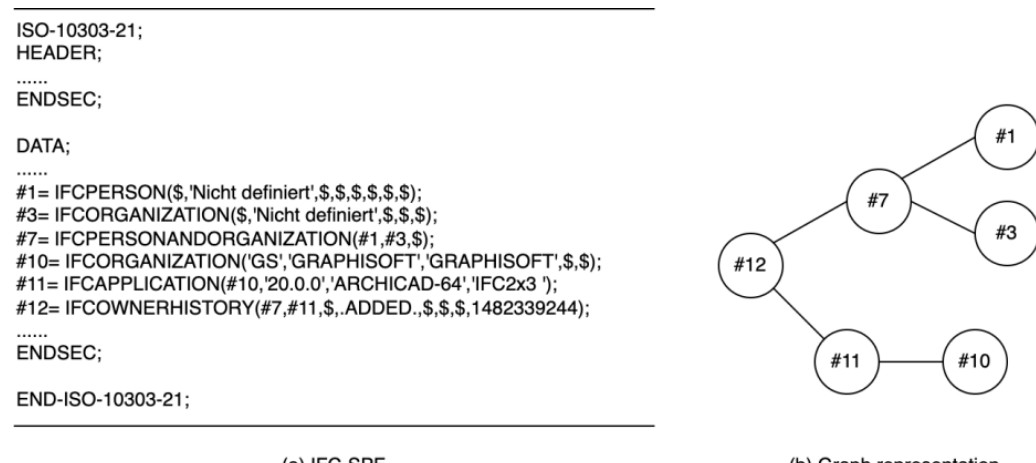

**Figure 5.** An example for part of IFC-SPF and its graph representation.

*4.2. The IFC-to-Graph Conversion*

4.2.1. Motivation for IFC-to-Graph

The conversion of IFC into a graph can bring many benefits, such as improved building information representation [43], clearer relationships between objects (see Figure 5), more effective data integration [66], improved interoperability [39,46], and improved information query and processing [67], they are the main motivations for converting IFC into a graph.

4.2.2. Related Works

Many studies are available on IFC-to-Graph conversion. For example, Khalili and Chua [68] studied the conversion of IFC into labelled graphs to facilitate topological queries on building elements. Tauscher et al. [67] developed a graph-based BIM query approach by converting the IFC object model into a graph. Gradišar and Dolenc [66] used a graph database (neo4j) to integrate IFC data with sensor data for monitoring bridge structural health. Gan [69] converted a subset of IFC defined in MVD (model view definition) into a graph data model to support automatic generative design for modular construction. Buruzs et al. [70] created an accessibility graph from IFC, which was used as an intermediate step for enriching building space information. Vilgertshofer and Borrmann [71] created a graph-based representation of product models, and graph transformation was used to create multi-scale versions of models for shield-tunnels, and Zhao et al. [72] converted IFC into a graph-based structure for merging IFC data.

4.2.3. Conversion Pattern

In terms of the conversion method, most studies adopted the pattern presented in Figure 6: (a) identifying information requirements in accordance with the need of the research project and designing a mapping strategy, (b) analysing IFC data structure to locate the required information and extracting the information, (c) converting the extracted IFC information into graphs using the mapping strategy designed in (a), and (d) applying graph algorithms to solve the research problem. This conversion pattern is common, but the specific conversion methods developed by these studies are mostly project specific.

4.2.4. Types of Graphs Identified

From the literature, it is noticed that IFC was mainly converted into two forms of graphs, i.e., RDF graphs in a triple store and property graphs in an LPG-based graph database. These two technologies are, however, developed for different purposes [50]. In general, the RDF technology is for the construction of the Semantic Web [73], while LPG is a graph model used by a native graph database [49]. By converting IFC into RDF, users expect to use the rule-based inference function, such as Pauwels et al. [74] and Ma et al. [75], and by converting IFC into LPG, users expect to use the graph-based analytic function, such

as Khalili and Chua [68], Tauscher et al. [67], and Skandhakumar et al. [76]. The next two sections present more details about RDF and LPG, as well as the IFC-to-LPG conversion and the IFC-to-RDF conversion.

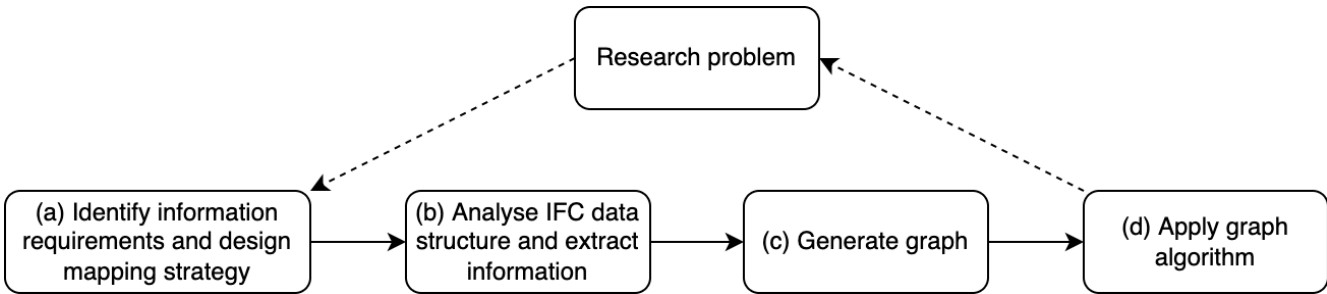

**Figure 6.** The basic pattern for IFC-to-graph conversion.

## 5. Labelled Property Graph and IFC-to-LPG Conversion

### 5.1. Database, NoSQL Database, and Graph Database

5.1.1. Relational Database and NoSQL Database

Strictly speaking, RDF and LPG are just data models, one uses the 'subject-predicate-object' structure and the other uses nodes and edges, which need to be implemented with database technology. In general, a database consists of two parts: a collection of data and a set of programs to access those data [14]. The conventional database is a relational database, such as MySQL [77], where data are stored in tables and queried using SQL (Structured Query Language).

In response to the rapid growth of data in velocity, variety and volume, NoSQL ('non SQL' or 'not only SQL') databases are developed to provide better performance in data storage, system scaling, and information query [78–80]. In general, NoSQL databases are those that do not use relational tables for data storage or use SQL for information queries. The common NoSQL databases include graph databases (e.g., neo4j and Amazon Neptune), key-value databases (e.g., Redis and DynamoDB), column family databases (or wide-column databases, such as Cassandra and HBase), and documents databases (e.g., MongoDB and CouchDB) [49,81]. It is, however, not a focus of this study to compare NoSQL databases with relational databases, nor compare graph databases with other types of NoSQL databases. For more detailed information about these databases, please refer to [14,49,82]. Instead, the focus of this study is mainly on the graph database.

5.1.2. Graph Database

According to Robinson et al. [49], a graph database is any database that, from the user's perspective, behaves like a graph database that exposes a graph data model through CRUD (create, read, update, and delete) operations. This definition of a graph database is subjective, inclusive, and thus broad. According to this definition, both LPG-based and RDF-based databases are graph databases. As with a relational database, a graph database provides full transactional database characteristics, including ACID (Atomicity, Consistency, Isolation, Durability) [32,83]. Moreover, compared with the relational database and many other NoSQL databases, a graph database is more efficient in representing relationships [49]. However, graph databases can be further divided into native graph databases and non-native graph databases, according to the mechanism used for data storage and processing. The relationships between different types of databases are summarised and presented in Figure 7, and the difference between native graph data and non-native graph databases is discussed in the next section.

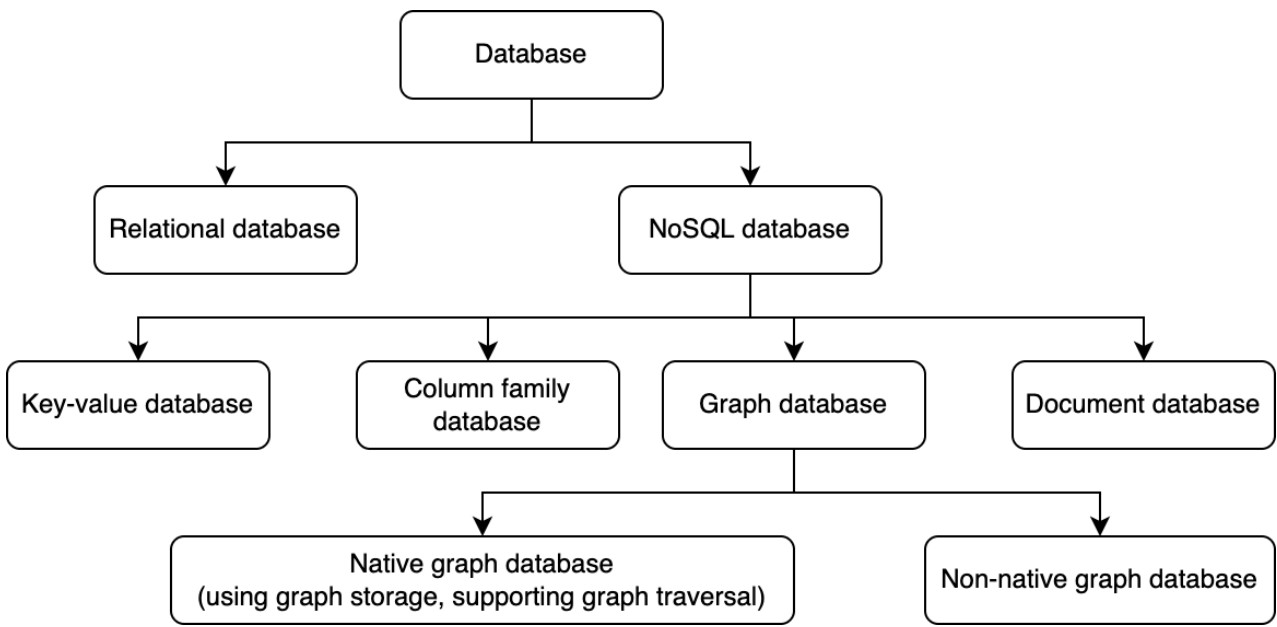

**Figure 7.** Types of databases and their relationship.

### 5.2. Native Graph Database and Non-Native Graph Database

In the category of a graph database, there are many available options on the market. According to G2.com [36], the top 10 graph databases are neo4j, ArangoDB, Amazon Neptune, Dgraph, DataStax, OrientDB, FlockDB, Cassandra, Titan, and Cayley. Among these databases, neo4j is leading in terms of database performance and market presence.

These different graph database systems can be distinguished from each other in two aspects, i.e., the underlying storage mechanism and the processing engine. In terms of storage, there are native graph storage and non-native graph storage, and in terms of the processing engine, there are native graph processing and non-native graph processing. Native graph storage enables performance and scalability, while native graph processing enables high-performance graph traversal by using index-free adjacency, which is the root of any graph-based algorithm [42]. Accordingly, graph databases can be divided into native graph database that uses native graph storage and native graph processing, and non-native graph database. The common native graph database includes neo4j, GraphBase, and TigerGraph [40]. In terms of conversion method, converting IFC into LPG (Section 5.3) is different from converting IFC into RDF (Section 6.3).

### 5.3. The IFC-to-LPG Conversion

5.3.1. The Motivation for IFC-to-LPG Conversion

LPG is a type of graph, in which nodes are labelled, edges are directed, and both nodes and edges have attributes [49]. Such a graph type can compactly store information within nodes and edges. In addition, LPG is compatible with other graph types (see Figure 8). By introducing certain constraints, LPG can be converted into other types of graphs. For example, removing properties from LPG will turn it into a labelled graph, which is the graph type for representing RDF [42]. As a result, LPG is the most commonly used graph type by native graph database systems [42,49], such as neo4j and Amazon Neptune [84]. Please note that if mentioned alone in this study, LPG refers to an LPG-based native graph database.

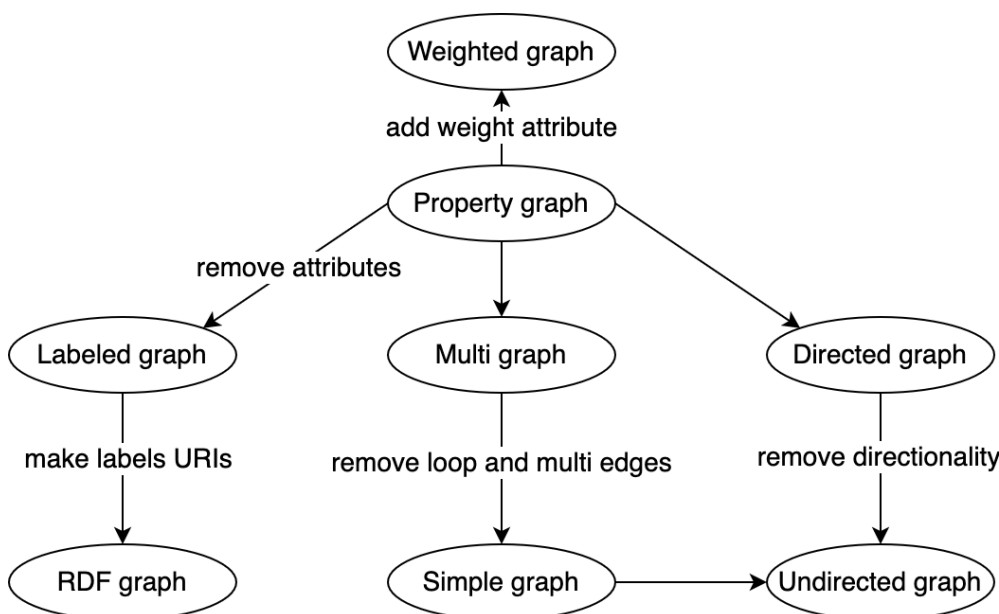

**Figure 8.** Types of graphs and their relationships.

Many studies investigated the conversion of IFC into LPG, such as studies by Ismail et al. [83,85], Hor et al. [32], Donkers et al. [37], Gradisar et al. [66], and Tauscher et al. [67]. Just like the purpose of converting IFC into other formats, such as shapefile and ifcOWL, the motivation for converting IFC into LPG is mainly to use the underlying analytical functions, such as Dijkstra's algorithm. For example, Nahar [86] used Dijkstra's algorithm to create emergency routes from IFC datasets. Tauscher et al. [67] used Dijkstra's algorithm to query the interlinked IFC information.

5.3.2. Challenges in IFC-to-LPG Conversion

The main challenges in IFC-to-LPG conversion come from two aspects, i.e., (a) concept mapping, or the mapping of instances, attributes to nodes and edges, and (b) the conversion method. A mapping process is required to decide how to create nodes and edges from instances, and attributes, while the conversion method determines the effectiveness and efficiency of the conversion.

(1)　Concept mapping

The IFC-to-LPG conversion mainly focuses on IFC data (i.e., a set of interrelated instances), instead of the IFC schema (i.e., a set of entities), and concerns the mapping of instances into nodes and edges (see Table 1) [76]. Most studies mapped the instances of object and property classes into the nodes and objects' direct attributes and nodes' properties.

**Table 1.** Concept mapping between IFC and LPG.

| IFC Concepts | LPG Concepts |
|---|---|
| Object | Node |
| Property | Node |
| Relationship | Node or edge |
| Attributes | Property or edge |

The main difference between these studies is in the mapping of relationship instances. Some studies, such as Ismail et al. [83,85], Nahar [86], and Zhao et al. [72], mapped the relationship instances to edges, while other studies, such as Tauscher and Crawford [87], mapped the relationship instances to nodes. This is largely dependent on the use case behind the conversion. For a building information query, the relationship instances should

be mapped to nodes, otherwise, there will be information loss [67,87]. Take the 'IfcRelSpace-Bounday' relationship [88], for example, this relationship has five attributes, excluding those inherited from parent classes. These attributes are helpful for creating indoor geometric networks (e.g., the 'PhyscialOrVirtualBounday' attribute) [76] and simplifying geometric building models (e.g., the 'InternalOrExternalBounday') [21]. These attributes will be omitted if relationship instances are converted into edges during the conversion, resulting in information loss.

(2)    The conversion method

The conversion methods used by studies are different. Most studies adopted a project-specific method and failed to address the problem in a more general way [89]. They extracted the required information from IFC and generated graphs by using the predefined mapping strategy. In these studies, only required information was extracted and converted, while other information was discarded. Ismail and Nahar [85,86] made the attempt towards a general-purpose IFC-to-LPG conversion method. However, the conversion method they developed is semi-automatic, as the to-be-converted IFC entities (classes) need to be manually listed in the conversion script, which makes their method not that practical, considering the large size of IFC, i.e., there are 816 entities in IFC4.2. Manually listing these entities, as well as their attributes, in the conversion script is labour-intensive and time-consuming, not to mention that the size of IFC is expected to grow in later versions. Figure 9. presents an example of the IFC-to-LPG conversion using an IfcProject instance. The original textual instance in IFC-SPF is presented in Figure 9a, the structure of graph database (neo4j) is in Figure 9b, and the LPG representation is in Figure 9c.

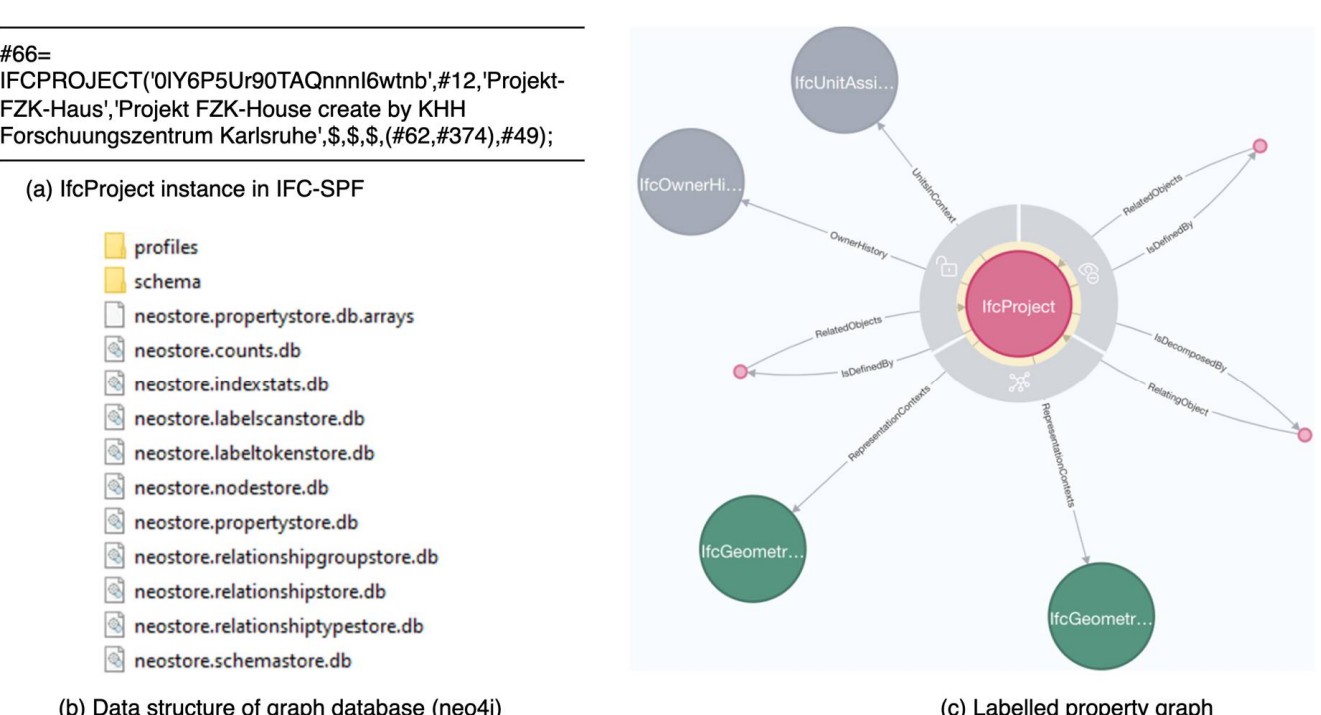

**Figure 9.** An example for the IFC-to-LPG conversion. (**a**) IfcProject instance in IFC-SPF, (**b**) data structure of graph database (neo4j), and (**c**) the LPG of the IfcProject instance.

## 6. RDF Graph and IFC-to-RDF Conversion

### 6.1. RDF and Semantic Web

RDF is closely related to the Semantic Web, which is an extension of the current Web and is referred to as linked data or the web of data [73,90]. The goal of the Semantic Web is to assist human users in everyday online activities [73]. Berners-Lee et al. [91] proposed the Semantic Web because humans were becoming less efficient in processing the daily

growing contents on the Web. They hoped that content on the web would be published in a way that can be understood by machines so that agents (or programs that collect Web content from diverse sources) can be developed to automatically collect and analyse information on the Web.

### 6.1.1. RDF for the Semantic Web

To realise this objective, data need to be published in a standard and common data model. Additionally, the data model should be domain-neutral [91], so that different areas can publish information using the same data model, regardless of the nature of the areas. RDF was developed for that purpose. The basic form of RDF, the 'subject-predicate-object' triple, can be used to represent information from various areas. Human users can then use RDF to publish data onto the Web and use SPARQL (SPARQL Protocol and RDF Query Language) to query RDF data. Another question concerning the development of the Semantic Web is how the machine can 'understand' the meaning (or semantics) of the data. This is realised by using ontologies described by ontology languages, such as the RDFS (RDF Schema) and OWL (Web Ontology Language) [73]. Eventually, intelligent agents can use the rules created by humans to automatically process online information, in which inferencing/reasoning is involved [73].

### 6.1.2. Relationships between Concepts in Semantic Web

The relationship among humans, machines (agent), RDF, rules, Linked Data, Semantic Web, and the conventional web is summarized and presented in Figure 10. The Semantic Web is the data infrastructure for automated machine-oriented data processing, it is based on the infrastructure of the conventional Web, which uses data from databases [73], including the conventional relational database or the relatively new NoSQL databases. RDF is used to describe both data and ontologies for the Semantic Web, SPARQL is used by human users for information queries, while rules are developed by humans for machines to conduct inferencing. Please note the following. (1) RDF data itself cannot do inferencing but supports inferencing, (2) RDF if mentioned alone in this paper, mainly refers to the RDF technology, including the 'subject-predicate-object' data model, syntax (such as Turtle and RDF/XML), and semantics (RDFS/OWL), which has the same meaning [73].

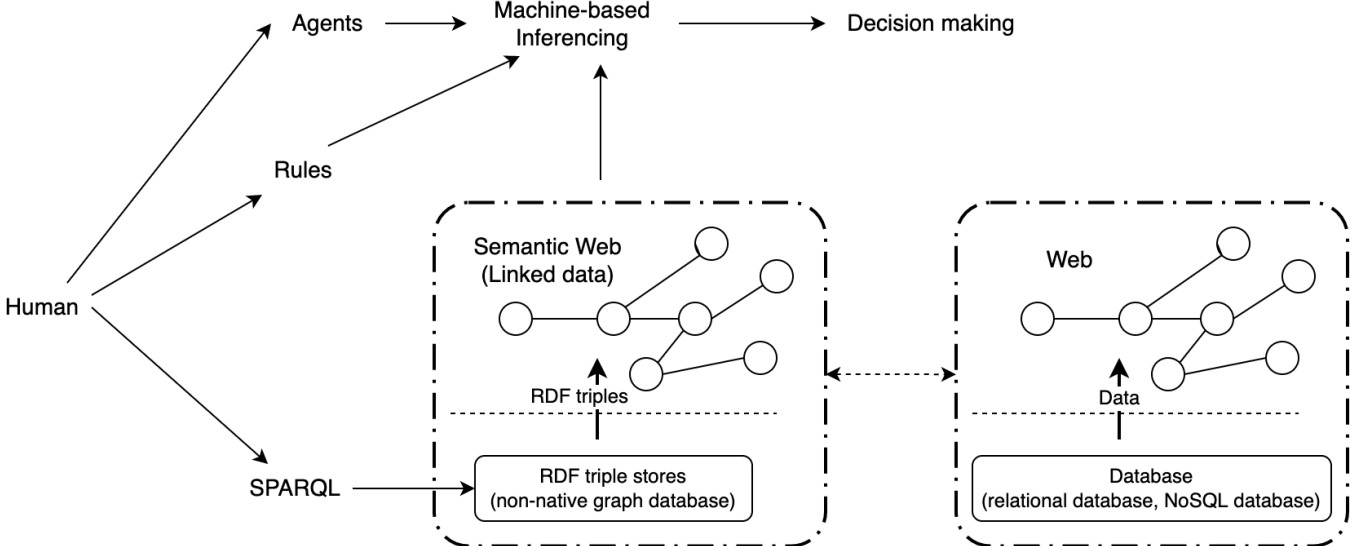

**Figure 10.** Relationship among humans, machine (agent), RDF, rules, Semantic Web, and Linked Data.

### 6.1.3. Semantic Web, Semantic Network, and Knowledge Graph

It should be noted that the 'Semantic Web' is different from the 'Semantic Network'. A semantic network (or just net) is a graphic notation for representing knowledge using

connected nodes and arcs, which was first developed for artificial intelligence and machine translation [92]. Another term, 'knowledge graph', is essentially the same as semantic networks [93], but with a difference in size, i.e., knowledge graphs are very large semantic networks [94]. It is important to distinguish these terms, as some studies mixed them up. The Semantic Web is in the domain of the Web [95], while semantic networks and knowledge graphs are in the domain of artificial intelligence [94].

### 6.2. RDF and RDF Graph

Even though the term 'RDF graph' is widely used by studies for describing RDF data, the core of RDF is not a graph, but the 'subject-predicate-object' triple. RDF triples can be visualised by using graphs in the form of nodes and directed-arc, according to W3C [45], after being processed by software, such as neo4j. In this sense, RDF graphs are labelled graphs (LG). The relationship between RDF triples and graphs is presented in Figure 11. The mapping between the 'subject-predicate-object' structure and the 'node-edge' structure is quite straightforward. Understanding this relationship would help distinguish RDF from LPG. RDF information represented using graphs can be interpreted and understood in an easier manner by humans. However, according to Antoniou et al. [73], this graphical syntax of RDF is not machine-interpretable, nor standardized. That is probably a reason why RDF triple stores are considered non-native graph databases [49].

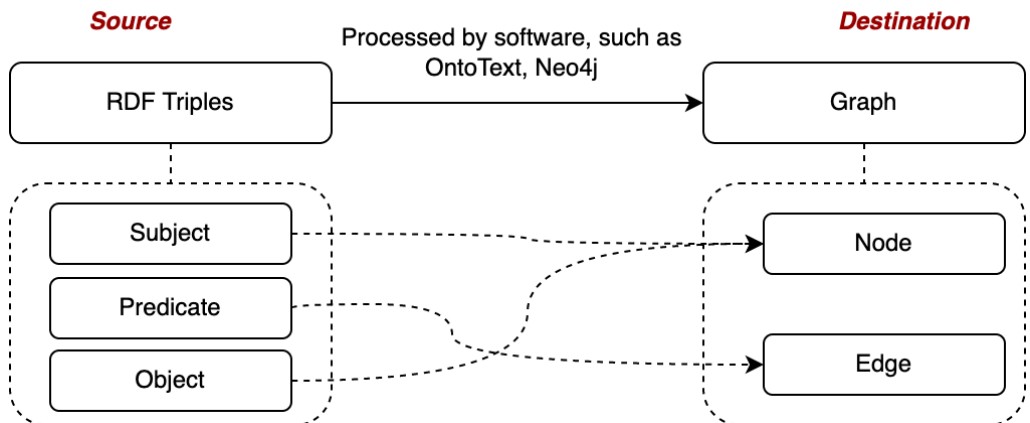

**Figure 11.** The relationship between RDF and graph.

### 6.3. IFC-to-RDF Conversion

6.3.1. The Motivation for IFC-to-RDF Conversion

Pauwels et al. [38] concluded three applications of the Semantic Web technology in the AEC domain, including improving interoperability, linking across a domain, and conducting logical inference. These three applications are closely related to the requirements and features of the Semantic Web. To be specific, to facilitate data publication, the data model (i.e., RDF) should provide high interoperability, and to facilitate information query for both humans and machines, the data on the Semantic Web should be interlinked, and finally, machine inference is used to process the published data [75]. These applications of the Semantic Web are the main motivations for the IFC-to-RDF conversion.

6.3.2. Challenges in IFC-to-RDF Conversion

Many studies have investigated the conversion of IFC into RDF. For example, Beetz et al. [96] initially investigated the conversion of the IFC EXPRESS schema to OWL using selected elements. Pauwels et al. [46,47] created a more comprehensive mapping between EXPRESS and OWL and developed a usable ifcOWL ontology, as well as an IFC-to-RDF converter [97]. To make the RDF datasets more compact and concise, Bonduel et al. [98] proposed the IFC-to-LBD (Linked Building Data) conversion, which was built on top of the LBD modular ontologies. Oraskari and Törmä [99] solved the

problem of assigning unique and stable identifiers to anonymous nodes in the RDF graph. Other studies on this topic include Yurchyshyna and Zarli [100], which converted IFC into RDF for conformance checking, and Torma [101], which proposed the Web of Building Data framework.

Different from the IFC-to-LPG conversion that focuses on IFC data and concerns the mapping of instances into nodes and edges, the IFC-to-RDF conversion focuses more on IFC schema and concerns the mapping of EXPRESS concepts to RDFS (RDF Schema) and OWL concepts, or how to 'describe' IFC schema by using concepts from RDFS and OWL with the 'subject-predicate-object' structure. For example, in the study by Beetz et al. [96], 'ENTITY' was mapped to 'owl:Class', 'SUBTYPE OF' and 'SUPERTYPE OF' were mapped to 'rdfs:subClassOf', and 'Attributes' was mapped to 'owl:ObjectProperty'. The IFC-to-RDF conversion also deals with IFC data, which is mainly a change of syntax [75]. Figure 12 presents an example of the IFC-to-RDF conversion by using an IfcProject instance (with a line number of #66).

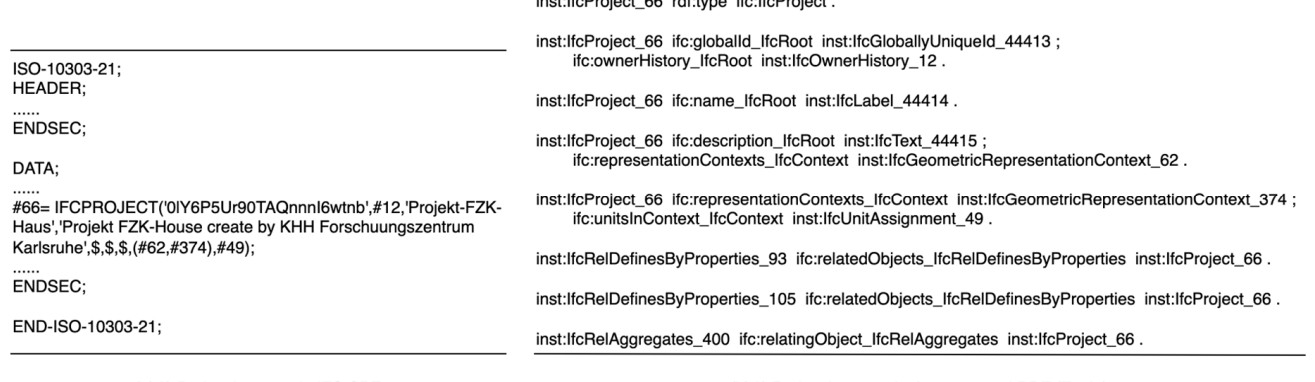

(a) IfcProject instance in IFC-SPF

(b) IfcProject instance in the converted RDF (Turtle)

**Figure 12.** An example for the IFC-to-RDF conversion.

### 6.4. RDF Information Query by Using SPARQL

RDF data can be queried by using SPARQL. The early version of SPARQL, however, did not have the fundamental navigational features (like pathfinding) [102]. There were some efforts trying to enable that feature for SPARQL, such as the nSPARQL by Perez et al. [102] and the SRelation by Mojžiš and Laclavik [103]. A new graph-like query technique, i.e., the property path, is included in the later SPARQL1.1, which allows pathfinding in the RDF graphs [104]. However, the mechanism behind the property path is different from graph traversal and this function is still problematic [103,105]. First, according to Arenas et al. [105], the property path is essentially a regular expression, which is a technique (not based on graph traversal) for extracting information from text [106]. The use of regular expressions is reasonable, as the machine-interpretable syntaxes of RDF, such as RDF/XML, Turtle, and RDFa [73], are all text-based. Second, this query mechanism is problematic and not often used. Arenas et al. [105] observed insufficient query performance when testing the property path function. Anyway, it is evident that SPARQL does not rely on graph traversal for querying RDF data, which is another reason why triple stores are not considered as a native graph database.

### 6.5. RDF-to-LPG Conversion

As mentioned above, RDF can be graphically represented by graphs, but the graphical representation cannot be interpreted by computers [42,73], and SPARQL-based operation has a limitation in implementing graph traversal algorithms [84]. That motivated studies to convert RDF into LPG, such as studies by Donkers et al. [37] and Hor et al. [32]. Donkers et al. [37] converted RDF datasets into LPG in neo4j to compare the query effi-

ciency of these two graph forms. Hor et al. [32] converted CityGML and IFC into RDF, which is then converted into LPG in neo4j.

The conversion of RDF into LPG is quite straightforward, subjects and objects are mapped to nodes, while predicates are mapped to edges (see Figure 11). This conversion has been supported by software applications, such as neo4j. Two plugins of neo4j, i.e., neosemantics (n10s) [107] and MSMNTX [108] can convert RDF datasets into LPG. After conversion, the labels (or URLs) of resources are turned into properties in nodes. Overall, the relationship between IFC, RDF, and LPG can be summarised in Figure 13. After the conversion, graph-based algorithms can be applied to the new LPG representation of RDF data. Figure 14 presents the LPG representation of RDF data in neo4j. It is obvious that the IFC-to-RDF conversion has changed the original IFC data structure.

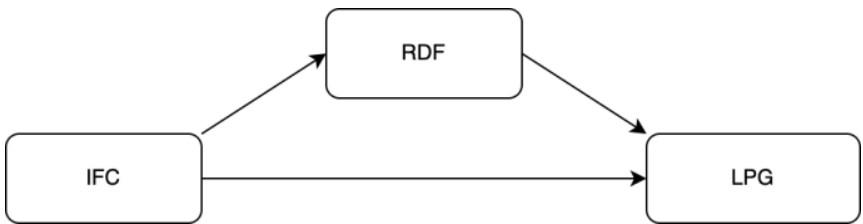

**Figure 13.** The pattern in IFC-to-Graph conversion.

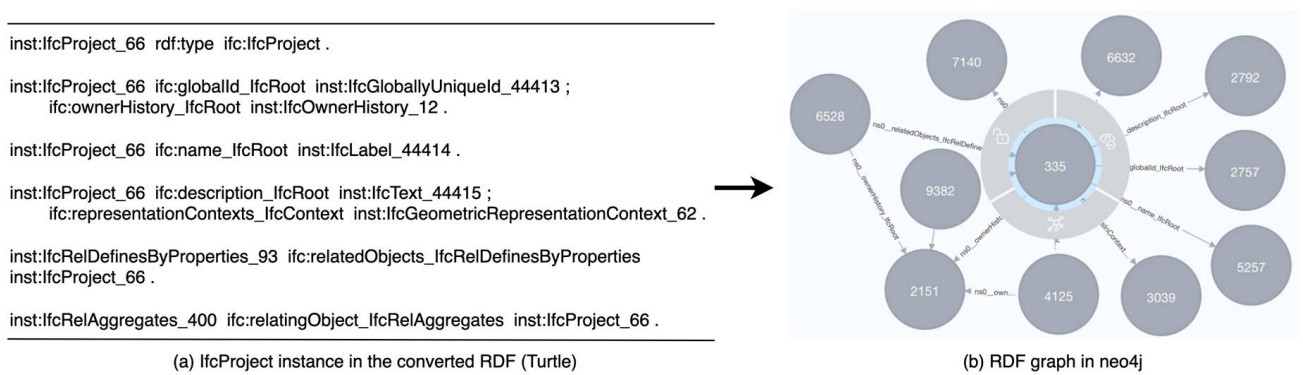

(a) IfcProject instance in the converted RDF (Turtle)　　　(b) RDF graph in neo4j

**Figure 14.** RDF data in (**a**) Turtle and (**b**) LPG representation.

## 7. The Graph for BIM/GIS Data Integration

### 7.1. The Requirements on Graph for BIM/GIS Data Integration

It is now clear that graphs can help address the interoperability issue between BIM and GIS, and we already have a clear understanding of RDF (triple store) and LPG (native graph database), the next question is which one is more suitable in BIM/GIS integration. The selection between RDF and LPG is largely dependent on the use case, as these two technologies have different performances in different scenarios. For example, LPG outperforms RDF in the context of linked data for smart homes [37], while RDF has a better performance in searching the substructure of glycan [41].

In the context of BIM/GIS integration, the selection between RDF and LPG is then dependent on the need for BIM/GIS data integration. BIM/GIS integration needs building information to be easily used in GIS, which requires the graph representation of IFC data to provide full building information access and an efficient information query. Full information access means semantic information in the original IFC data (e.g., IFC-SPF) should be fully accessed by GIS, and efficient information query requires information to be easily retrieved, which depends on two aspects, i.e., an expressive and powerful query language and a proper underlying data model.

Discussions on developing a query language are beyond the scope of this study; the focus is instead on a proper underlying graph data model. The underlying data model is

important and should be carefully designed, as it decides how information can be queried. When creating the ifcOWL ontology, Pauwels and Terkaj [46] proposed three criteria: (1) the ifcOWL ontology must be in OWL2 DL, (2) ifcOWL ontology should match the original EXPRESS schema as closely as possible, and (3) the ifcOWL ontology should primarily aim at the conversion of IFC instance files. Even though these three criteria were designed for the IFC-to-RDF conversion, the last two criteria still apply here in a broader context of IFC-to-graph conversion, namely (1) the graph data model should match the original IFC data model as closely as possible, or ideally, a full representation of the IFC data model, and (2) the focus should be on the conversion of IFC instance files. In addition to these two criteria, the graph data model should also be compact to provide storage efficiency and support easy information access and effective information query.

In summary, to facilitate BIM/GIS data integration, the graph should be (1) ideally a full representation of IFC (semantic) data, (2) as close as possible to the original IFC data model, (3) compact, and (4) supporting easy information access and effective information query.

*7.2. Comparing LPG with RDF*

Table 2 presents the comparison between the LPG technology (representing the native graph database) and the RDF technology (representing the Semantic Web).

**Table 2.** Comparison between LPG and RDF.

| Needs of BIM/GIS Data Integration | LPG/Native Graph Database | RDF/Semantic Web |
|---|---|---|
| Representing IFC data | Yes | Yes |
| Data model close to the original model | Yes | No |
| Compact data model | Yes | No |
| Easy information access | Yes | Yes |
| Efficient information query | Yes | Yes |
| Native graph storage | Yes | No |
| Native graph processing | Yes | No |

(1) Expressiveness. Expressiveness is the capability of LPG and RDF in representing IFC data. Both LPG and RDF can represent IFC data. LPG uses nodes, edges, as well as their properties to represent IFC data, while RDF can represent IFC data by using the 'subject-predicate-object' data structure.

(2) Adherence to IFC. LPG can retain the original IFC data model, while RDF has to change the original IFC data model to suit its 'subject-predicate-object' structure. Instances in IFC data can be represented by nodes, and their attributes can be stored as properties of nodes, or edges. Overall, LPG can fully respect the original IFC data model. In contrast, RDF actually 'redescribes' the IFC data by using its own 'subject-predicate-object' data structure, which inevitably changes the original IFC data structure. This change may prevent users from using this technique, due to the additional efforts required to learn a new data model.

(3) Data compactness. LPG is a compact data model, while RDF is not. RDF breaks everything down into minimum units (see Figure 12), this feature of RDF is required by the Semantic Web, which needs a domain-neutral data model for publishing heterogeneous data from different areas. Overall, RDF mainly concerns simplicity, universality, and flexibility [109], but not compactness. However, compactness should be a concern for graph databases that aim to minimise storage. That is why there are efforts trying to make RDF more compact, such as Hartig [110].

(4) Information access and query. Both LPG and RDF support easy information access via database technology. Data stored in these two forms can be accessed easily via free and common database API. In terms of information query, due to the use of a compact graph data model, which represents the original IFC data model, LPG can also provide a more efficient information query than RDF. First, graph-based

information query using index-free adjacency is faster than traditional index-based information query [49]. Second, a full LPG representation of IFC data can facilitate information queries, as users do not have to learn a new data model.

(5) Data security and shareability. The Semantic Web encourages users to publicly publish and share information on the Web so that the information can be linked, searched, and used by other Web users or agents [91]. That is why 5-star data on the Semantic Web must be 'open data' [90]. Therefore, the success of the Semantic Web will largely depend on open data and users' willingness to share and publish data using RDF. However, in the context of BIM/GIS integration, not all building information can be published and publicly shared, especially confidential information. Please note that it is possible to control the access to RDF data by using the database's authorization functionality [14] so that only authorised users can have access, but that is against the original purpose of RDF, nor in the favour of the Semantic Web that encourages publishing and sharing data.

### 7.3. The Chosen Graph

Overall, as mentioned earlier, the motivation behind the data conversion is to use the analytical functions/algorithms that the destination formats support. For IFC-to-RDF conversion, users expect to use the inferencing function, and for IFC-to-LPG, users expect to use the graph algorithms. The relationship between these types of conversion is summarized and presented in Figure 15.

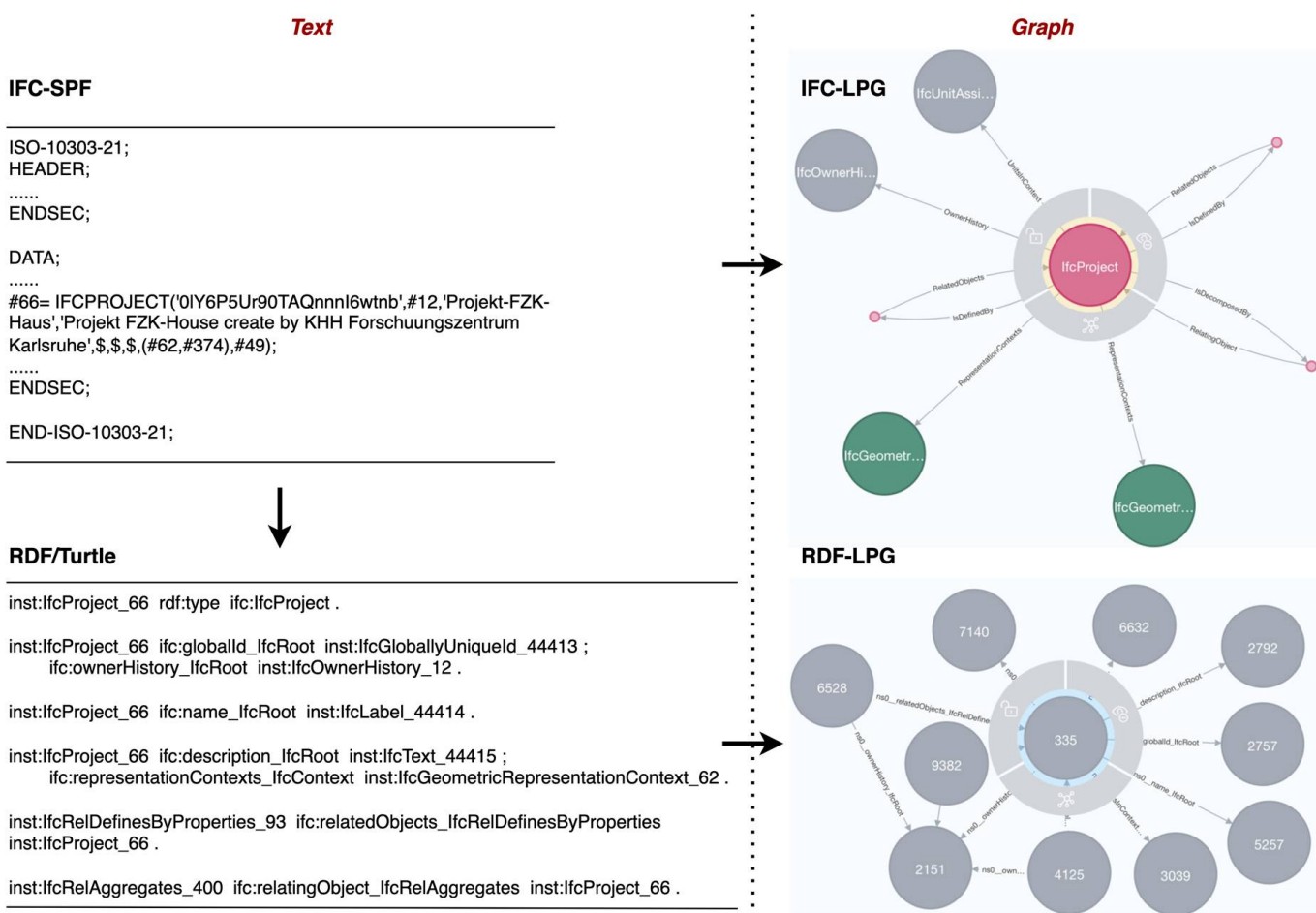

**Figure 15.** The conversion of IFC-SPF into RDF, IFC-LPG, and RDF-LPG.

In terms of application, RDF would suit applications where information needs to be publicly shared and published, and LPG would suit applications where graph algorithms are to be used [42]. In terms of function, RDF is mainly used for publishing, linking, and exchanging data over the Web, while LPG is used for data storage and querying [37]. From the perspective of time, LPG is a technology for the near future, it is adopted by a growing number of companies, such as HP, Amazon, and Cisco [111], competing with conventional relational databases, while RDF is for the far future when online information (resources) can be automatically processed by machines and used in decision making. Overall, considering the criteria, LPG is considered more suitable than RDF in the context of BIM/GIS integration.

It should be noted that many researchers tried to introduce the Semantic Web technology into the AEC domain [46,74,75], hoping to use the inference function to reveal the hidden relationships within building information, but many of them overlooked the fact that this reasoning/inferencing function is not automatic but requires human intervention. Human users develop rules for machines by using rule languages, such as the Semantic Web Rule Language (SWRL) [112], to enable them to infer like humans [37]. This programming process is, however, very similar to querying linked data by using Cypher [49].

### 7.4. Selection between RDF and LPG in the AEC Domain

Even though this study focuses on selecting a proper graph technology for BIM/GIS integration, the methodology presented in this study would also benefit other studies that aim to introduce new technology into the AEC domain.

Many studies tried to introduce new technologies into the AEC domain but failed to first conduct a systematic investigation. An example is the use of CityGML in BIM/GIS integration. It appears that initial studies selected CityGML mainly because it is an international standard used in the geospatial industry, while its initial purpose and the features of the geospatial industry were not sufficiently investigated. They ignored the fact that the surface-model based CityGML suits the geospatial industry because laser scanning and photogrammetry are commonly used in this industry for collecting surface information, which is then used for generating digital terrain [113,114] or buildings [115–117]. Converting the solid-model based IFC data into CityGML requires a solid-to-surface conversion, which was and still is problematic [24], while the geospatial industry can actually use solid models in formats such as shapefile and GML.

In the case of RDF, the situation is similar. Many studies adopted RDF because of its high interoperability and the inference/reasoning function it supports, without investigating its original purpose for the Semantic Web and the features of the Semantic Web that have been described by Berners-Lee et al. [91]. In this sense, the systematic investigation of RDF and LPG, which involves their initial purpose, features, and functions, would benefit studies in the AEC domain that intend to use these two technologies.

## 8. Future Work on Graph towards More Effective BIM/GIS Data Integration

The use of LPG will bring a new pattern of using semantic building information in GIS, as presented in Figure 16, which can complement the system-oriented paths or dataset-oriented paths [24]. The benefit of this new pattern is that problematic class mapping will no longer be required.

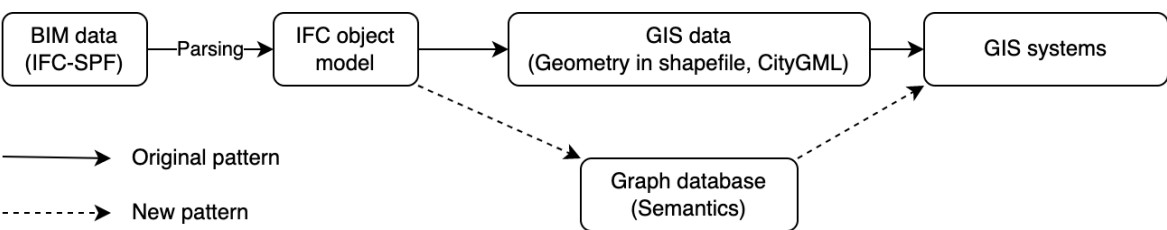

**Figure 16.** The new pattern for using building information in GIS.

To create graphs for more effective BIM/GIS data integration, the following aspects should be investigated in the future, including (1) an efficient way for generating a graph from IFC data, (2) a proper data model for IFC graph data, and (3) tailored queries for IFC-based graph data, and (4) properly handling geometric building information.

### 8.1. An Efficient Way for Generating LPG Graph

Most of the current methods for IFC-to-LPG conversion are project-specific, a more generic and fully automatic conversion is required. Despite the clear requirements, there is currently no effective way to create full LPG for IFC data, this is mainly because, as mentioned above, most studies were focused on using the graph algorithms to solve their problems, rather than representing IFC data [89]. The proposed graph generation method should be generic and fully automatic [83,85] and can fully represent the semantic information in the original IFC data.

We are currently working on this aspect, trying to develop a method that is fully automatic and can fully convert IFC data into LPG to support effective and efficient building information access and query, and the preliminary outcome is promising. With a valid method for fully converting IFC into LPG, it will also be possible to conduct a further comprehensive comparison between RDF and LPG by using experiments.

### 8.2. A Proper Data Model for IFC Data

While LPG can fully represent the original IFC data model, the full graph representation might not be the best form for information query [67], for two reasons. (a) The graph is ineffective in representing geometric information, as the graph cannot represent ordered lists. (b) Some attributes in IFC, such as the 'OwnerHistory' attribute [118], can hinder effective information queries.

IFC uses an object-oriented model, where child classes (or entities) can inherit attributes from parent classes, and IfcRoot is the root class for most entities in the IFC standard. As a result, most entities have the 'OwnerHistory' attribute and most instances in the IFC data file would reference the same IfcOwnerHistory instance. This would cause problems in finding the shortest paths to studies similar to Ismail et al. [85]. This problem is common for both LPG and RDF. To illustrate this problem, an IFC-SPF of a house model from IFC Wiki [119] was converted into RDF by using the tool developed by Pauwels [97], which was then imported into neo4j by using the neosemantics (n10s) plugin [107]. Cypher was used to query nodes that are directly linked to the IfcOwnerHistory object using the following query "match (n)–(m) where n.uri CONTAINS 'IfcOwnerHistory' return n, m". The query result is presented in Figure 17, which shows that all 82 building components (walls, windows, doors, etc.) of the house model and other instances of IfcRoot are directly linked to the IfcOwnerHistory object. The consequence is that in most cases the shortest path between two nodes within the graph would be Object1 → IfcOwnerHistory → Object 2.

This problem, however, has not been recognised by researchers investigating RDF. None of those major studies on IFC-to-RDF conversion discussed or mentioned this issue, probably because RDF triple stores do not rely on graph traversal. However, this problem needs to be properly addressed for LPG.

### 8.3. Tailored Queries for IFC-Based Graph Data

When a fully automatic conversion method and a proper data model are available, the next concern is regarding the query language, which is another aspect related to efficient and effective information queries.

There are many languages available on the market for querying LPG, such as Cypher for neo4j [120], openCypher [121], GraphQL [122], and Gremlin [123], there is even a new ISO standard under development for graph query language (GQL) [124]. These graph query languages are for general purposes, not specific to building data. The performance of the graph query is determined by the size of the graph traversed [68] or the expressiveness of the query. If a query is over-expressive, more nodes will be traversed, resulting in longer

query time (or lower efficiency). Domain knowledge should be incorporated to refine queries, which can improve query performance [125,126].

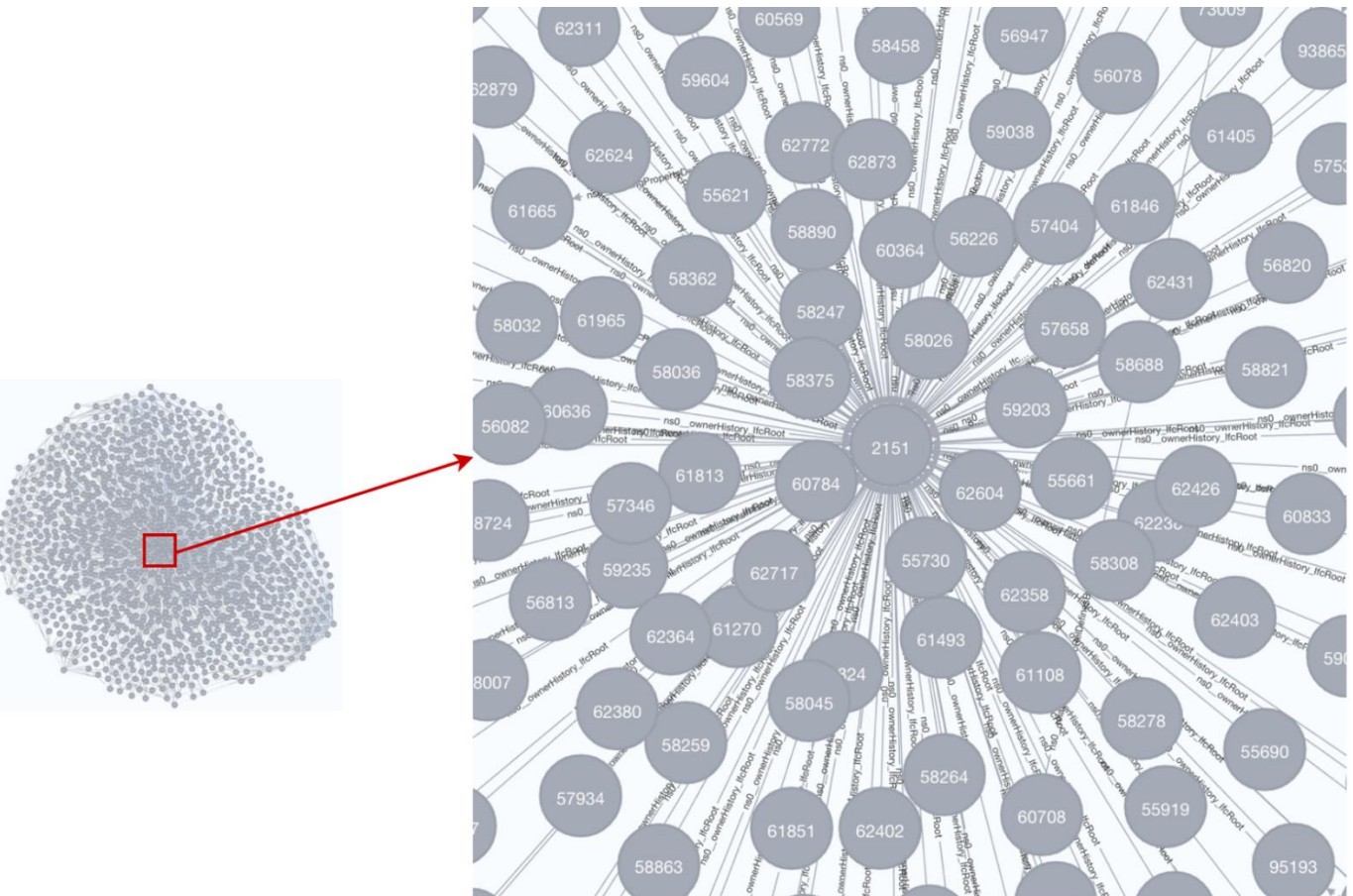

**Figure 17.** The shortest path problem in representing IFC data using graph.

### 8.4. Handling Geometric Building Information

BIM information generally consists of two parts, i.e., geometry and semantics. While graphs are effective in representing semantic BIM information, they may have a problem in representing geometric information. For instance, Pauwels et al. [47] noticed that graphs are not effective in representing ordered lists, as the order of nodes is not a native concept in graph theory [87].

The geometry of BIM models generally falls into two types, i.e., explicit models and implicit models [26]. On one hand, implicit models mainly refer to swept solid, CSG (Constructive Solid Geometry), and clipping, which are represented by a set of parameters, whereas the final shapes will only be calculated when needed from those parameters. On the other hand, explicit models are those represented by boundary representation (B-rep), where the shape has been created and every point in the shape is explicit. Examples of implicit model and explicit model are presented in Figure 18 using the same shape, i.e., a cubic with a size of $1 \times 1 \times 1$.

When it comes to handling geometric information using a graph, two questions need to be properly addressed. The first is if the geometric information should be stored in a graph and if yes, the second question is then how to store geometric information in LPG, especially B-rep.

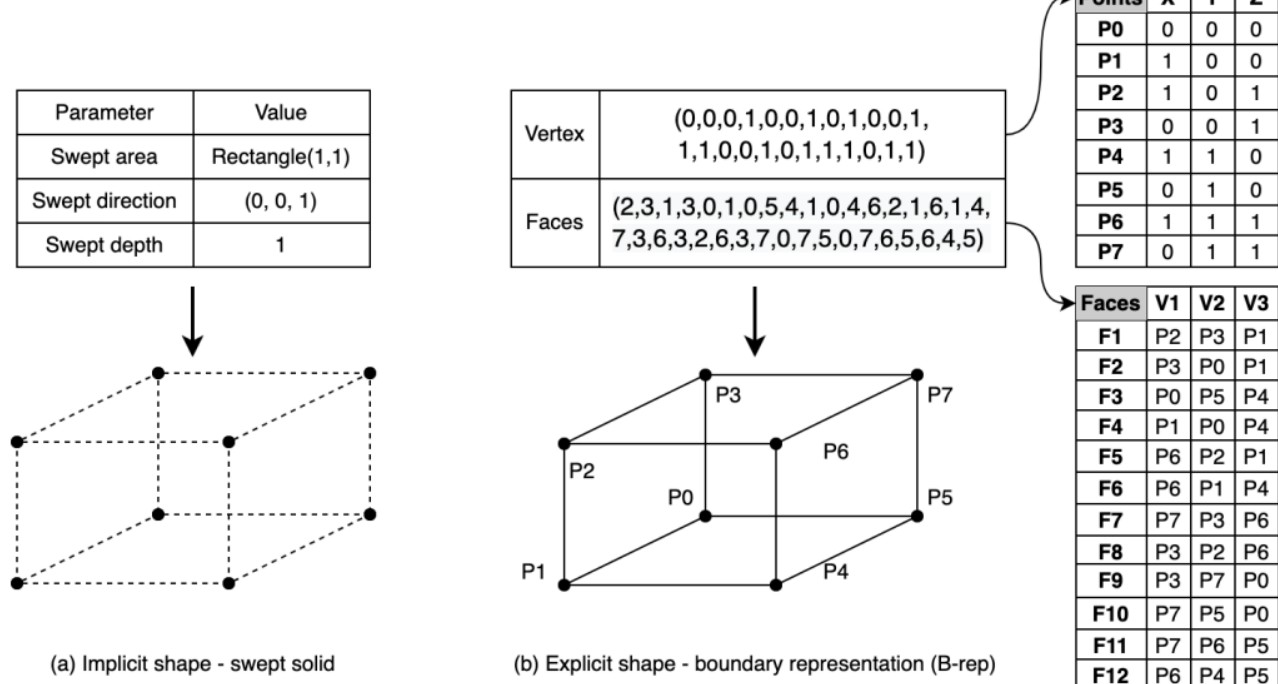

**Figure 18.** Examples of (**a**) implicit model and (**b**) explicit model.

(1) For the first question, there is currently no strong motivation to store geometric information in a graph. First, there is no graph-based geometry engine available for interpreting geometric information stored in graphs, which means the shape cannot be directly calculated from the graph by using existing tools. Second, there is no strong use case. It is unlikely for users to make queries like 'what is the location of the first point in the first window instance?'. In this sense, the geometric information can be excluded from the graph and managed by using other formats, such as shapefile. A similar situation has been discussed in [127] for RDF.

(2) Regardless of the above concerns, if the answer to the first question is still yes, a proper way for storing geometric information in a graph is required. There are two potential solutions. (a) In the first solution, implicit models can be directly stored in the graph as they are and for explicit models, the solution proposed by Pauwels [47] can be adopted to enable the graph to represent ordered lists. (b) The second solution is to convert shapes into plain text [127] and store that text as a property of nodes. The benefit of the second solution is twofold. First, the shape information in plain text can be extracted and reused by existing geometry engines, such as OCCT (Open CASCADE Technology) [128]. Second, the graph size (i.e., the number of edges [129]) will not be significantly increased.

Overall, in the context of BIM/GIS integration, currently, a practical solution is to exclude geometry from the graph, as most studies managed the geometric information and semantic information [24,26] separately and the geometric information can be managed in a more efficient way outside the graph.

## 9. Conclusions

This paper was motivated by the following four questions. (1) How can graph technology contribute to BIM/GIS data integration? (2) What is the relationship between RDF and graphs, while RDF is described as a graph-based data model, graph-based analysis methods are seldom applied to RDF datasets. (3) What is the relationship between RDF and LPG, while both are considered graph-based data models? (4) Which technology is more suitable for BIM/GIS data integration, the RDF-based triple store, or the LPG-based

native graph database? To answer these questions, this paper reviewed recent papers and relevant books on this topic and conducted a systematic comparison. The main findings and outcomes of this paper are as follows.

There is a limited number of studies on the direct use of graphs in BIM/GIS data integration, but there are more studies available investigating the representation of IFC data using graphs, which can contribute to the early steps of BIM-to-GIS data conversion. Moreover, most studies converted IFC data into RDF graphs or LPG graphs that are stored in graph databases. A graph database is a type of NoSQL database, the counterpart of which is the conventional relational database.

Both RDF and LPG can use a graph to represent building information and improve the interoperability of building data, but they are designed for different purposes. A brief summary is presented in Table 3.

**Table 3.** Summary for RDF and LPG.

|  | **RDF** | **LPG** |
|---|---|---|
| Original purpose | Linking data for Semantic Web | A graph type for native graph database |
| Strength | Information publishing | Data query |
| Conversion of IFC | IFC to 'subject-predicate-object' | IFC to nodes and edges |
| Graph database | Non-native graph database | Native graph database |
| The use of graph | Visualisation | Data query and visualisation |

- RDF and LPG are two essentially different technologies and were created for different purposes. RDF is initially designed for linking heterogeneous global data to support the construction of the Semantic Web, while LPG is a graph type adopted by a native graph database.
- IFC can be converted into RDF for information publishing, sharing, and linking; IFC can also be converted into LPG for more efficient and effective data query, supported by a graph traversal algorithm. These two conversion purposes do not conflict with each other. However, these two types of conversions have different concerns during the conversion. IFC-to-RDF converts IFC data into a 'subject-predicate-object' structure, while IFC-to-LPG converts IFC data into nodes and edges.
- In a broader sense, RDF triple stores can be considered graph databases, because they deal with data that is logically linked, but they are not native graph databases, as they are not optimised for graph storage and do not support index-free adjacency for efficient graph traversal.
- The information query mechanism of SPARQL is different from a graph. The core of RDF is the 'subject-predicate-object' triple; a graph is just a way of visualisation and is not machine-interpretable. To utilise graph algorithms, RDF triples need to be further converted into nodes and edges.
- In general, RDF suits applications where data sharing and linking are the focus, while LPG suits applications where graph algorithms are to be used.

A set of requirements on the graph for BIM/GIS integration are identified. The graph should first be (a) ideally a full representation of IFC (semantic) data, (b) compact enough to provide storage efficiency, (c) as close as possible to the original IFC data model, and (d) supporting easy information access and effective information query.

Based on these requirements, LPG is thought to be more suitable than RDF for BIM/GIS data integration. However, the following aspects should be improved in the future to facilitate the use of graphs in BIM/GIS integration, including (a) an efficient way for generating a graph from IFC data, (b) a proper graph data model for IFC data to support efficient information query, and (c) tailored queries for IFC-based graph data, and (d) properly handling geometric building information.

With the adoption of graphs in BIM/GIS data integration, it is expected to bring a change to the information use pattern on the GIS side, which would allow semantic

building information to be more efficiently and effectively used. Overall, the systematic investigation into RDF and LPG would also be beneficial to other studies in the AEC domain that intend to use these two technologies. Understanding the difference between these two technologies would help choose the proper technology, while both are generally considered graph-based technology. This study has presented a preliminary theoretical comparison between LPG and RDF; however, a further comparison using BIM models should also be conducted when an approach for fully converting IFC into LPG is available.

**Author Contributions:** Conceptualization, J.Z., H.-Y.C., H.Z., J.W. and Y.T.; methodology, J.Z. and H.-Y.C.; software, J.Z.; formal analysis, J.Z.; data curation, J.Z.; writing—original draft preparation, J.Z.; writing—review and editing, J.Z. and H.X.; funding acquisition, H.X. All authors have read and agreed to the published version of the manuscript.

**Funding:** This research was funded by the Australian Research Council, grant number LP160100528; Curtin University, grant number CUB 2022; Curtin University, grant number SRG2022.

**Institutional Review Board Statement:** Not applicable.

**Informed Consent Statement:** Not applicable.

**Data Availability Statement:** Not applicable.

**Acknowledgments:** The authors would like to thank the anonymous reviewers for their comments and suggestions that helped improve the comprehensiveness and clarity of our paper.

**Conflicts of Interest:** The authors declare no conflict of interest.

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
