# Peer review of "The Application of Graph in BIM/GIS Integration"

_buildings, doi:10.3390/buildings12122162_

Round 1

Reviewer 1 Report (Previous Reviewer 4)

Dear Author

The paper is very good and well-presented. Please add these to the article to improve it. 

1. Please extend section 1.1

2. Figure 9, 12, 14, 15. Can we improve the quality of these figures? 

3. Add missing details of references

4. Can you add a final result table and present the key findings of the methods in a table for ease? 

Author Response

Reviewer 2 Report (New Reviewer)

The paper draws attention to an interesting subject. The paper provides valuable information. The overall level of the paper is good. However, I have just a few small comments on the manuscript.

The main title of the study should be clearer. Please improve the main title of the study.

Figure resolution/quality needs to be improved.

It may be better to highlight the limitations of the study. This would strengthen the discussion and the conclusions of the paper.

It may be better to give a list of abbreviations.

In the manuscript, there are a few typographical errors. Please check it.

Author Response

Reviewer 3 Report (New Reviewer)

When it comes to identifying a need to review the literature on a certain topic of interest, it must be clearly established to 'Why' such a need was felt and the focus of the work should be accordingly maintained? What it contributes towards industry and academia and audience in general? What are the key take aways and recommendations?

Also it's not clear what specifically "systematic investigation" means in the abstract, i.e. what it intends to achieve in light of the above?

While many relevant and similar studies have been conducted in the past (as follows), why authors have not cited these works. Also the authors of present review article have not done comparative analysis from such similar studies. That is it would be better to categorise paper based on application aspects.

Elsheikh et al. 2021: https://doi.org/10.1088/1757-899X/1090/1/012128

Khan 2022: https://doi.org/10.1007/978-981-19-2145-2_17

Song et al. 2017: https://doi.org/10.3390/ijgi6120397

And what's is different from previous publication? :

https://doi.org/10.1080/19475683.2020.1743355

Available software packages (such as AutoDesk, https://damassets.autodesk.net/content/dam/autodesk/www/pdfs/autodesk-bring-together-bim-gis-ebook-final.pdf) already offer seemless BIM-GIS integration, so their advantages or disadvantages (over other studies) have been left out?

Please explain the need or advantage of such a review when more advanced work is available? such as :

https://doi.org/10.1007/s11831-021-09545-2

Line 39: Reference no. [9] (a review article) is wrongly placed and can be moved elsewhere.

Line 88: References [28-29] are bundled together, what are the differences between them? It would be better to expand the a little.

Lines 824-826: What does the observation in the last sentence intend to mean? Do you mean laboratory or field experiments on real construction projects? How such experiments will be designed for comparison?

Author Response

Reviewer 4 Report (New Reviewer)

Overall, it is a profound, well-organized, and well-written study. I would like to highlight minor discrepancies for the improvement of the manuscript.

1)      I would advise the author to add the significance of this study in the ABSTRACT.

2)      In the INTRODUCTION, there is no connectivity between the starting two sentences, i.e., the first and second sentences (lines 29 to 34), reflecting different contexts. Kindly create some connective background.

3)      In the INTRODUCTION, the following sentences are either incomplete or the reference is not proper (Author [ref]).

“ A thorough review on BIM/GIS data integration has been presented in [25].” (line 92-93)

“The interconnected BIM information, which can be thought of as a graph [22],” (line 106)

4)      It is suggested to make METHODOLOGY a separate section, i.e., section 2, rather than making it part of INTRODUCTION (section 1.7).

5)      Figures 9, 12, 14, & 15 are either blurred or invisible; kindly improve them.

6)      I would suggest the author to write the conclusion in paragraph form rather than numbering and bullets.

Author Response

Reviewer 5 Report (New Reviewer)

Thank you for the opportunity to review this article. Overall, it is a very well-structured, well-written and very interesting article.

I have one minor and one major comment:

Minor: In clause 1.6, line 138, You start with "Apart from RDF..." although RDF has not been introduced yet. Please rewrite this sentence, or present RDF before the sentence. 

Except for this minor change, the subclauses on motivation and methodology are excellent. 

Major: In the comparison of RDF and LPG in clause 6, you present table 2. Based on the comparison in table 2, you then conclude in subclause 6.3 (line 635) and clause 8 (line 812) that you consider LPG more suitable than RDF for BIM/GIS integration. I feel that you have too little fundament to come to this conclusion. In table 2, you have not considered, for example, the openness of the two technologies. LPG seems more locked to specific database systems, while RDF is a standardized technology that can be accessed independently of database systems and even as standardized file serialization. Also, RDF/OWL ontologies are part of an RDF graph, which is essential for querying and understanding the information. Furthermore, you have not considered the standardization of technologies and query languages, where RDF has advantages with RDF, RDFS, OWL and SPARQL. These issues are essential for information access.

Finally, you have not considered the flexibility of graphs in the two technologies. Using properties in LPG may be challenging for extending the graph from those properties. I have also seen research describing challenges with handling complex structures (which IFC is an example of) in LPG. See, for example, Jelinek, Skoda and Hoksza, 2017 and Alocci et al., 2015. 

Therefore, I question whether your conclusion is too absolute. The picture is more complex than you have presented, and you should include other perspectives, like those mentioned here.

Round 2

Reviewer 5 Report (New Reviewer)

Thank you for the thorough feedback. Although I do not fully agree with your answers to my comments, I find this article to be ready for publication now. I look forward to future documentation of your further work on this relevant and essential topic. 

Author Response

This manuscript is a resubmission of an earlier submission. The following is a list of the peer review reports and author responses from that submission.

Round 1

Reviewer 1 Report

Well-organized paper with a sound approach. A few minor issues could improve the message conveyed and the readability of the paper, eg on Figure 1 it would be desirable to indicate the specifics of the interoperability between systems A and B, as it is - it looks redundant, in addition, the visibility of Figure 18 should be improved, so that it attracts readers attention, maybe zoom-in of one portion would be appropriate. 

Reviewer 2 Report

The studies focus on technological problems and solutions for the integration of BIM and GIS technologies.

The article is applicable to MDPI's Informatics or Technologies journals. Not compatible with Buildings.

The literature review is not compatible with the introduction since the article did not research anything about Smart Cities nor how the data contributes to the Digital Twins and the keywords are also not compatible with the cited themes.

The article is only applicable to journals in the information technology area not having connections with the Buildings standard.

Reviewer 3 Report

v

The issue of BIM-GIS interoperability is very topical and of fundamental importance for the construction sector. The paper presents a robust analysis, supported by data and results compatible with the objectives. For this reviewer the paper can be published as is, being sufficiently complete in all its parts.

Reviewer 4 Report

Dear Authors 

The manuscript is good and presented well. Figure 1 and 18 are not clear. Please add more details about them. Though, the concept is good but I am not sure about the validity of the suggested approach. It would be good if you add it.